# Prediction of Postoperative Pulmonary Edema Risk Using Machine Learning

**DOI:** 10.3390/jcm12051804

**Published:** 2023-02-23

**Authors:** Jong Ho Kim, Youngmi Kim, Kookhyun Yoo, Minguan Kim, Seong Sik Kang, Young-Suk Kwon, Jae Jun Lee

**Affiliations:** 1Department of Anesthesiology and Pain Medicine, Chuncheon Sacred Heart Hospital, Hallym University College of Medicine, Chuncheon-si 24253, Republic of Korea; 2Institute of New Frontier Research Team, Hallym University College of Medicine, Chuncheon-si 24252, Republic of Korea; 3Department of Anesthesiology and Pain Medicine, College of Medicine, Kangwon National University, Chuncheon-si 24341, Republic of Korea

**Keywords:** lung, edema, surgery, prediction, machine learning

## Abstract

Postoperative pulmonary edema (PPE) is a well-known postoperative complication. We hypothesized that a machine learning model could predict PPE risk using pre- and intraoperative data, thereby improving postoperative management. This retrospective study analyzed the medical records of patients aged > 18 years who underwent surgery between January 2011 and November 2021 at five South Korean hospitals. Data from four hospitals (*n* = 221,908) were used as the training dataset, whereas data from the remaining hospital (*n* = 34,991) were used as the test dataset. The machine learning algorithms used were extreme gradient boosting, light-gradient boosting machine, multilayer perceptron, logistic regression, and balanced random forest (BRF). The prediction abilities of the machine learning models were assessed using the area under the receiver operating characteristic curve, feature importance, and average precisions of precision-recall curve, precision, recall, f1 score, and accuracy. PPE occurred in 3584 (1.6%) and 1896 (5.4%) patients in the training and test sets, respectively. The BRF model exhibited the best performance (area under the receiver operating characteristic curve: 0.91, 95% confidence interval: 0.84–0.98). However, its precision and f1 score metrics were not good. The five major features included arterial line monitoring, American Society of Anesthesiologists physical status, urine output, age, and Foley catheter status. Machine learning models (e.g., BRF) could predict PPE risk and improve clinical decision-making, thereby enhancing postoperative management.

## 1. Introduction

Postoperative pulmonary edema (PPE) is a well-known complication with multiple possible causes [1]. Preexisting cardiac disease, including heart failure, is the most common cause of PPE. Fluid overload results in increased hydrostatic pressure and worsening left ventricular function [2]. Regardless of preexisting heart disease, fluid overload itself can cause PPE. In particular, excessive postoperative fluid administration and transfusions increase the risk of PPE [1,3]. Neurogenic pulmonary edema is another potential cause of PPE [4]. Although neurogenic pulmonary edema is sometimes regarded as a form of acute respiratory distress syndrome, its pathophysiology and prognosis differ from the characteristics of acute respiratory distress [4,5]. PPE can also be caused by anaphylaxis, which results in negative pressure and acute lung injury [1,6].

It is often difficult to determine the cause of PPE during its early stages, particularly in patients with overlapping etiologies [1,6,7]. There is a need to identify patients at high risk of PPE to allow prevention and early treatment. Several studies have reported the causes and risk factors for PPE, but early diagnosis and management are difficult, more so in patients with overlapping etiologies or uncertain causes [1,2,3,4,5,6,7,8,9,10].

Advances in computing have enhanced several key areas of clinical research; artificial-intelligence-based methods may have additional applications. Machine learning (ML) systems are widely used in clinical research to analyze big data. Compared to traditional scoring systems, ML models perform better when predicting various clinical conditions [11,12,13]. They have been successfully used to predict postoperative complications [14,15,16,17,18,19]. However, there is no reported ML model to predict PPE. In the present study, we hypothesized that ML could predict PPE risk with good performance, and then developed ML models to predict PPE.

## 2. Materials and Methods

### 2.1. Data Collection

This retrospective cohort study protocol was approved by the Clinical Research Ethics Committee of Chuncheon Sacred Heart Hospital, Hallym University. The need for informed consent was waived because of the retrospective study design. The medical records of patients treated between 1 January 2011 and 15 November 2021 were obtained from the clinical data warehouses of five hospitals affiliated with Hallym University Medical Center. The hospitals were located in Seoul (Kangnam Sacred Heart Hospital and Hangang Sacred Heart Hospital), Gyeonggi Province (Hallym University Sacred Heart Hospital and Dongtan Sacred Heart Hospital), and Gangwon Province (Chuncheon Sacred Heart Hospital).

A clinical data warehouse is a database of medical records, prescriptions, and test results, which can be used to identify patients based on prescriptions, examinations, and diagnostic data. The timing and results of investigations, drug administration, transfusions, and other information were extracted in an unstructured text format. The requested data were provided in a de-identified format, but the data of specific patients could be extracted using a key.

### 2.2. Patients and Pulmonary Edema

The study included adult patients aged > 18 years who did not exhibit preoperative pulmonary edema. The exclusion criteria and outlier data were missing. Pulmonary edema was diagnosed by radiologists on the basis of chest radiographs. Patients were presumed not to have PPE if they lacked perioperative respiratory symptoms and did not undergo chest radiography.

### 2.3. Dataset

The dataset involved the following 98 perioperative variables: age, male sex, and order of surgery; the statuses of preoperative atelectasis, preoperative effusion, preoperative pneumothorax, preoperative pneumonia, preoperative pulmonary thromboembolism, and preoperative acute respiratory distress; body mass index; the statuses of congestive heart failure, cardiac arrhythmia, valvular diseases, pulmonary circulation disorders, peripheral vascular disorders, hypertension (uncomplicated vs. complicated), paralysis, other neurological disorders, chronic pulmonary diseases, diabetes (uncomplicated vs. complicated), hypothyroidism, renal failure, liver diseases, peptic ulcer diseases (excluding bleeding), acquired immune deficiency syndrome/human immunodeficiency virus, lymphoma, metastatic cancer, solid tumors (without metastasis), and rheumatoid arthritis/collagen vascular diseases; alcohol consumption, current smoking status, smoking frequency (packs), smoking duration (years), emergency status, American Society of Anesthesiologists physical status of >2, use of general anesthesia, maintenance anesthetics administered, N_2_O use, anesthesia time (min), surgery time (min), intraoperative blood and fluid administration, intraoperative urine output, and estimated blood loss; the statuses of arterial line monitoring, central venous pressure monitoring, Foley catheter, Levin tube, and patient-controlled analgesia; the administration of intraoperative packed red blood cells, frozen fresh plasma, platelets (concentration and cryoprecipitate), rocuronium, vecuronium, atracurium, cisatracurium, succinylcholine, pyridostigmine, neostigmine, sugammadex, fentanyl, alfentanil, sufentanil, remifentanil, and pethidine; blood urea nitrogen level, creatinine level, glomerular filtration rate, prothrombin time, activated partial thromboplastin time, and platelet count; the levels of sodium, potassium, uric acid, protein, and albumin; and the statuses of robotic surgery, laparoscopic surgery, heart surgery, abdominal surgery, breast surgery, ear surgery, endocrine surgery, eye surgery, head and neck surgery, musculoskeletal surgery, neurosurgery, obstetric and gynecological surgery, spine surgery, thoracic surgery, transplant surgery, urogenital surgery, vascular surgery, and skin and soft tissue surgery.

The dataset was divided into training and test sets. The training set included data from Kangnam Sacred Heart Hospital, Hangang Sacred Heart Hospital, Hallym University Sacred Heart Hospital, and Dongtan Sacred Heart Hospital. The test set included data from Chuncheon Sacred Heart Hospital. The training set was used for model learning, whereas the test set was used to evaluate model performance. Both datasets were standardized using min.–max. scaling based on the training set.

### 2.4. Machine Learning

The study used supervised learning, which is an ML paradigm for data consisting of labeled examples (i.e., each data point contains variables and an associated label). Five ML algorithms were used: random forest, light-gradient boosting machine, extreme-gradient boosting machine, multilayer perceptron, and logistic regression [20,21,22,23,24]. Random forest is a regression tree technique that uses bootstrap aggregation and predictor randomization to achieve high predictive accuracy. Various random forest input parameters were explored [25]. A light-gradient boosting machine continuously divides a leaf node with maximum data loss without a consideration of tree balance, resulting in a deep and asymmetric tree [26]. Extreme-gradient boosting machine is an optimized gradient boosting algorithm that involves parallel processing, tree-pruning, missing value management, and regularization to avoid overfitting/bias [27]. Multilayer perceptron is a neural network with ≥1 intermediate layer between the input and output layers. The network is connected in the direction of the input, hidden, and output layers; there are no connections within the layers, but the output layer is directly connected to the input layer through a feedforward network [28]. Logistic regression can solve the binary classification problems associated with the linear model.

The dataset was imbalanced and may have caused low model performance. Therefore, we used the synthetic minority oversampling technique for all algorithms except random forest [29]. After the ratio of pulmonary edema had been balanced, we trained the models with a training set that included synthetic samples. The random forest algorithm includes a classifier method known as balanced random forest (BRF); therefore, the synthetic minority oversampling technique was not used for the random forest algorithm. Data processing and the ML process are summarized in Figure 1. Feature importance was calculated to assess the best model using the built-in function in the algorithm package.

### 2.5. Modified Dataset

An additional carved dataset was used to modify the prediction model based on the large and complex dataset. This dataset was learned and validated using the best prediction algorithm from the original data. First, the test dataset was reduced by under-sampling using the Tomek’s link method to validate our best model [30]. Second, a simplified prediction model was made using 20 important features of the best model, and was validated using a test dataset that included these features.

### 2.6. Metrics and Statistics

Six metrics were calculated for model performance. The primary metric was the area under the receiver operating characteristic curve. The average precisions of precision-recall curve, best threshold, precision, recall and f1 score, and accuracy were calculated. Google Colab (Python version 3.7; Google, Mountain View, CA, USA) was used to calculate model metrics.

Descriptive analysis was performed to compare the characteristics of patients with and without PPE. Categorical variables were presented as numbers (%) and compared using the chi-squared test. Continuous variables were presented as medians (interquartile ranges) and compared using the Mann–Whitney U test. *p*-values of < 0.05 were considered statistically significant.

## 3. Results

### 3.1. Patient Characteristics

The study included 287,976 patients aged > 18 years who did not exhibit preoperative pulmonary edema. After the exclusion of 26,597 patients with missing (*n* = 26,593) and outlier (*n* = 4) data, and 4480 preoperative PPE patients, a total of 256,899 patients were included in the analysis. PPE occurred in 5480 (2.8%) patients. The training and test sets included 221,908 and 34,991 patients, respectively. PPE occurred in 3584 (2.1%) and 1896 (7.4%) patients in the training and test sets, respectively (Table 1 and Table 2).

### 3.2. Model Performance

BRF exhibited the best performance for the prediction of PPE risk. As the primary metric, the area under the receiver operating characteristic curve for BRF was 0.91 (95% confidence interval: 0.84–0.98). The performances of the remaining models are summarized in Figure 2. BRF also exhibited the best performance based on the average precision of the precision-recall curve (0.44). The average precisions of the precision-recall curve for the remaining models are summarized in Figure 3. BRF had the best recall (0.832) and f1 score (0.372), whereas the light-gradient boosting machine model had the best precision (0.531) and accuracy (0.946). The remaining metrics are summarized in Table 3.

### 3.3. Feature Importance

The evaluation of feature importance in the BRF model revealed that arterial line monitoring was the most important feature. Ten major features in the BRF model are shown in Figure 4.

### 3.4. Validation of under-Sampling Test Dataset and Simplified Model

After under-sampling of the test dataset, PPE patients were 1896 and No-PPE patients were 32,621. In the simplified prediction model, the included features were as follows: arterial monitoring, American Society of Anesthesiologists physical status, age, urine output, intraoperative fluid, estimated blood loss, foley catheter, anesthesia time, albumin, glomerular filtration rate, central venous pressure monitoring, operation time, prothrombin time, blood urea nitrogen, protein, creatinine, prothrombin time-international normalized ratio, platelet, body mass index, and intraoperative packed red blood cell. Validation results are summarized in Table 4.

## 4. Discussion

We used ML to develop models for the prediction of PPE. Model training using data from 221,908 patients was followed by model testing using data from 34,991 patients. Five algorithms were used to develop the models, whereas six metrics were used to evaluate their performances. BRF exhibited the best performance in terms of area under the receiver operating characteristic curve, recall, and accuracy. However, no model had a good precision or f1 score.

Numerous studies have developed ML models to predict postoperative pulmonary complications. Peng et al. developed and validated a deep-neural-network model based on combined natural language data and structured data to predict pulmonary complications in geriatric patients [15]. Xue et al. developed an ML model to predict postoperative pulmonary complications after emergency gastrointestinal surgery in patients with acute diffuse peritonitis [18]. Chen and colleagues developed an ML model to predict postoperative pneumonia in orthotopic liver transplant patients [14]. Although the outcomes of the above studies included PPE, their findings differed from ours because they also assessed other complications. An ML model to predict PPE risk after any type of surgery has not been developed.

PPE has various causes, several of which can occur simultaneously. PPE may be cardiogenic or noncardiogenic, but it is difficult to distinguish between these etiologies because of their similar clinical features. In patients with acute myocardial infarction, cardiogenic pulmonary edema may be complicated by noncardiogenic edema related to the aspiration of gastric contents, syncope, or cardiac arrest. Conversely, in patients with severe trauma or infections accompanied by noncardiogenic pulmonary edema, fluid resuscitation may cause pulmonary edema through volume overload and increased pulmonary vascular hydrostatic pressure [1,6,31]. Therefore, PPE prediction and the preemptive management of risk factors are important.

The present study investigated the important features of the best model for the prediction of PPE risk. Ten major PPE risk factors were included, primarily those related to fluid and hydrostatic pressure rather than the other causes of PPE. This means that the PPE prediction model could mainly predict cardiogenic and hydrostatic pulmonary edema. However, the evidence is not conclusive because the etiologies of PPE in this study were not known.

The most important feature was arterial line monitoring, which is required in patients who need continuous blood pressure monitoring or multiple blood sampling during surgery [32]. Arterial line monitoring is the standard of care for patients at risk of rapid hemodynamic changes. Patients with a poor preoperative status and those who undergo major surgeries can develop rapid hemodynamic changes and often need multiple sampling [33]. American Society of Anesthesiologists physical status and age also indicate preoperative patient condition. Patients with high American Society of Anesthesiologists physical status grades may develop heart, lung, kidney, and brain problems [34,35]. Old age is generally associated with compromised organ function, resulting in a greater risk of PPE [36]. Urine output, fluid volume, EBL, albumin, and glomerular filtration rate directly and indirectly affect body fluid status, which is associated with hydrostatic pressure [37,38,39,40,41].

To the best of our knowledge, our model is the first to predict PPE risk, and its performance was better than the previous PPE models. However, the present study had some limitations. First, the overall performance of the model was good, but its precision and f1 scores were low, even for the best threshold. Because recall (sensitivity) was good, the proportion of false positives may be high, presumably because of the low proportion of patients with pulmonary edema in the overall dataset. Thus, our model interpreted the normal state as PPE in many cases. There were similar results in the validation with the under-sampling dataset. Second, our model requires many features to predict PPE, which reduces its practicality. Although the performance was not significantly worse in the model with twenty features, this limitation of the model could not be resolved. A prediction model based on fewer features while maintaining the performance may be needed in the future. To resolve the first two limitations, additional datasets should be acquired and learned, or features with better predictive values should be selected. Third, our model could not distinguish between cardiogenic and noncardiogenic PPE. Additional studies are needed to develop models that can distinguish between the two PPE types and predict the risk of each type.

In conclusion, we developed an ML model that could predict PPE risk in patients undergoing surgery. The model was superior to previously reported prediction models for postoperative pulmonary complications. Our ML model may improve clinical decision-making, thereby enhancing postoperative management. However, further improvements are needed to reduce the false positive rate and enhance the practical usefulness.

## Figures and Tables

**Figure 1 jcm-12-01804-f001:**
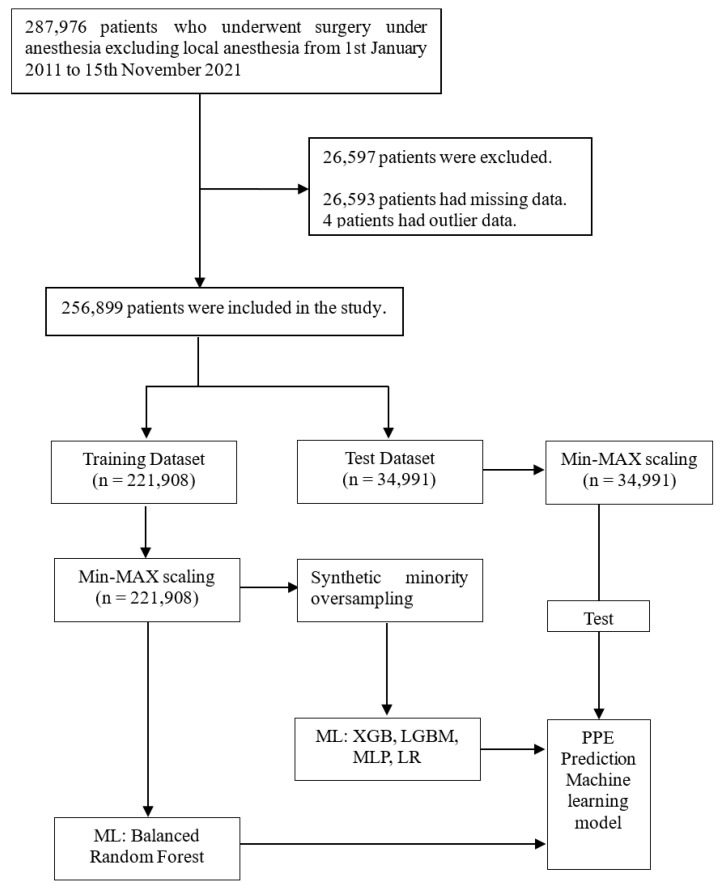
Data processing and machine learning process. LGBM, light-gradient boosting machine; LR, logistic regression; ML, machine learning; MLP, multilayer perceptron; XGB, extreme-gradient boosting machine.

**Figure 2 jcm-12-01804-f002:**
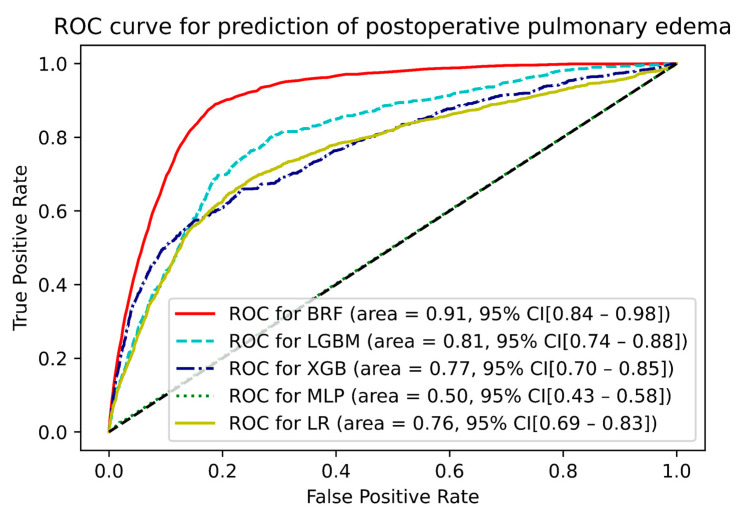
Receiver operating characteristic curve for prediction of postoperative pulmonary edema risk. BRF, balanced random forest; CI, confidence interval; LGBM, light-gradient boosting machine; LR, logistic regression; MLP, multilayer perceptron; ROC, receiver operating characteristic; XGB, extreme-gradient boosting machine.

**Figure 3 jcm-12-01804-f003:**
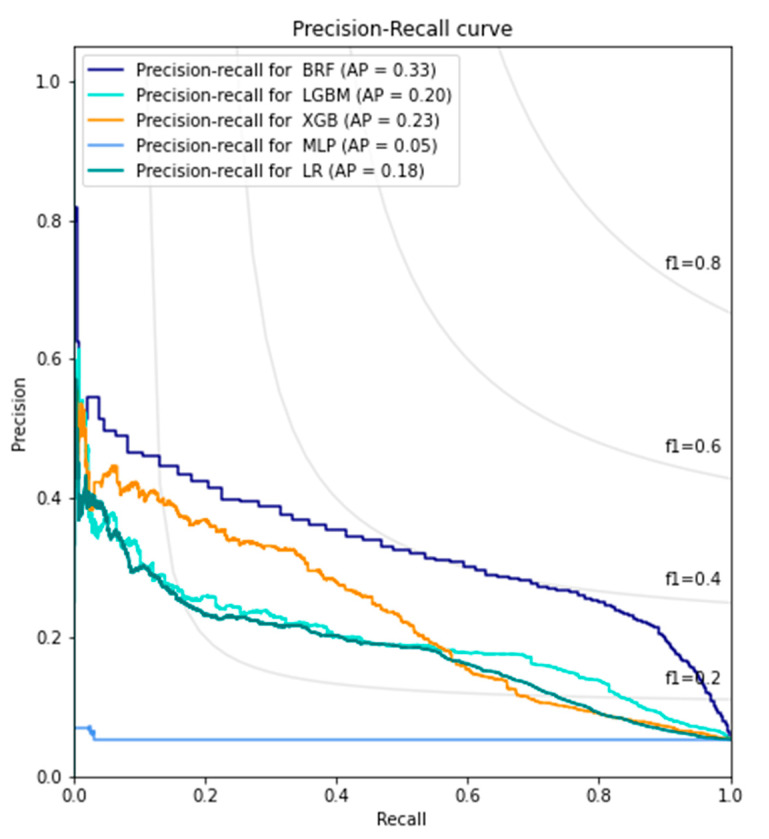
Precision-recall curve for prediction of postoperative pulmonary edema risk. BRF, balanced random forest; CI, confidence interval; LGBM, light-gradient boosting machine; LR, logistic regression; MLP, multilayer perceptron; AP, average precision; ROC, receiver operating characteristic; XGB, extreme-gradient boosting machine.

**Figure 4 jcm-12-01804-f004:**
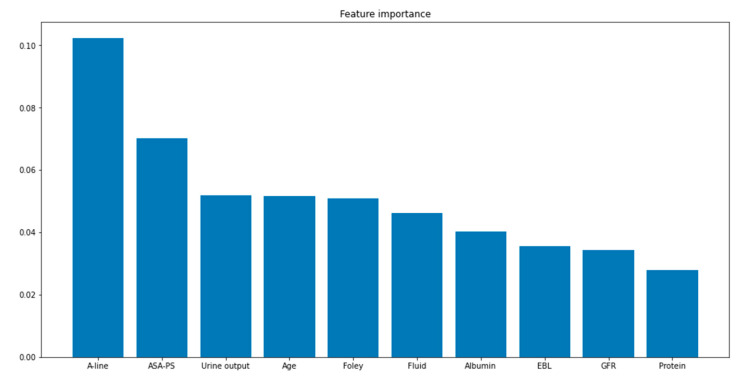
Ten major features of the balanced random forest model for prediction of postoperative pulmonary edema risk. A-line, arterial line monitoring; ASA-PS, American Society of Anesthesiologists physical status; EBL, estimated blood loss; Fluid, fluid administration; Foley, Foley catheter use; GFR, glomerular filtration rate; Protein, protein level.

**Table 1 jcm-12-01804-t001:** Training dataset.

	No PPE (*n* = 218,324)	PPE (*n* = 3584)	*p*-Value
Age	50.0 (37.0, 62.0)	69.0 (56.0, 78.0)	<0.001
Male sex	100,552 (46.1)	1739 (48.5)	0.004
Order of surgery	1.0 (1.0, 1.0)	1.0 (1.0, 1.0)	<0.001
Preoperative atelectasis	6276 (2.9)	418 (11.7)	<0.001
Preoperative effusion	4658 (2.1)	410 (11.4)	<0.001
Preoperative pneumothorax	1968 (0.9)	71 (2.0)	<0.001
Preoperative pneumonia	1401 (0.6)	176 (4.9)	<0.001
Preoperative PTE	178 (0.1)	18 (0.5)	<0.001
Preoperative ARDS	19 (0.0)	1 (0.0)	0.754
Body mass index	24.0 (21.8, 26.5)	23.9 (21.5, 26.6)	0.059
Congestive heart failure	4236 (1.9)	467 (13.0)	<0.001
Cardiac arrhythmia	6520 (3.0)	353 (9.8)	<0.001
Valvular diseases	1170 (0.5)	153 (4.3)	<0.001
Pulmonary circulation disorders	921 (0.4)	77 (2.1)	<0.001
Peripheral vascular disorders	4399 (2.0)	192 (5.4)	<0.001
Hypertension, uncomplicated	24,413 (11.2)	903 (25.2)	<0.001
Hypertension, complicated	9025 (4.1)	420 (11.7)	<0.001
Paralysis	609 (0.3)	33 (0.9)	<0.001
Other neurological disorders	5120 (2.4)	243 (6.8)	<0.001
Chronic pulmonary diseases	14,365 (6.6)	394 (11.0)	<0.001
Diabetes, uncomplicated	11,051 (5.1)	373 (10.4)	<0.001
Diabetes, complicated	14,996 (6.9)	562 (15.7)	<0.001
Hypothyroidism	4828 (2.2)	162 (4.5)	<0.001
Renal failure	5847 (2.7)	426 (11.9)	<0.001
Liver disease	11,293 (5.2)	244 (6.8)	<0.001
Peptic ulcer diseases (excluding bleeding)	4055 (1.9)	92 (2.6)	0.002
AIDS/HIV	59 (0.0)	0 (0)	0.640
Lymphoma	744 (0.3)	26 (0.7)	<0.001
Metastatic cancer	2261 (1.0)	77 (2.1)	<0.001
Solid tumors without metastasis	33,956 (15.6)	991 (27.6)	<0.001
Rheumatoid arthritis/collagen vascular diseases	3176 (1.4)	83 (2.3)	<0.001
Alcohol consumption	59,560 (27.3)	698 (19.5)	<0.001
Current smoking	37,885 (17.4)	533 (14.9)	<0.001
Smoking frequency (packs)	0.0 (0.0, 0.0)	0.0 (0.0, 0.0)	<0.001
Smoking duration (years)	0.0 (0.0, 0.0)	0.0 (0.0, 0.0)	0.014
Emergency	36,571 (16.8)	1057 (29.5)	<0.001
ASA-PS > 2	82,608 (37.8)	1053 (0.5)	<0.001
General anesthesia	181,268 (83.0)	3210 (89.6)	<0.001
Anesthetics (sevoflurane)	124,935 (57.2)	2415 (1.1)	<0.001
N_2_O	10,226 (4.7)	73 (2.0)	<0.001
Anesthesia time (min)	105.0 (70.0, 160.0)	190.0 (125.0, 300.0)	<0.001
Surgery time (min)	65.0 (35.0, 115.0)	130.0 (75.0, 230.5)	<0.001
Intraoperative blood administration	0.0 (0.0, 0.0)	0.0 (0.0, 400.0)	<0.001
Intraoperative fluid administration	500.0 (300.0, 900.0)	1500.0 (850.0, 2500.0)	<0.001
Intraoperative urine output	0.0 (0.0, 90.0)	215.0 (60.0, 550.0)	<0.001
Estimated blood loss	20.0 (0.0, 100.0)	500.0 (100.0, 800.0)	<0.001
Arterial line	56,990 (26.1)	3252 (90.7)	<0.001
Central venous line	14,362 (6.6)	1649 (46.0)	<0.001
Foley catheter	77,786 (35.6)	3097 (86.4)	<0.001
Levin tube	3656 (1.7)	277 (7.7)	<0.001
Patient-controlled analgesia (intravenous)	83,711 (38.3)	1119 (0.5)	<0.001
Intraoperative packed red blood cells	0.0 (0.0, 0.0)	0.0 (0.0, 2.0)	<0.001
Intraoperative FFP	0.0 (0.0, 0.0)	0.0 (0.0, 0.0)	<0.001
Intraoperative PC	0.0 (0.0, 0.0)	0.0 (0.0, 0.0)	<0.001
Intraoperative cryoprecipitate	0.0 (0.0, 0.0)	0.0 (0.0, 0.0)	<0.001
Rocuronium	50.0 (0.0, 50.0)	50.0 (25.0, 60.0)	<0.001
Vecuronium	0.0 (0.0, 0.0)	0.0 (0.0, 0.0)	<0.001
Atracurium	0.0 (0.0, 0.0)	0.0 (0.0, 0.0)	0.214
Cisatracurium	0.0 (0.0, 0.0)	0.0 (0.0, 0.0)	<0.001
Succinylcholine	0.0 (0.0, 0.0)	0.0 (0.0, 0.0)	<0.001
Pyridostigmine	0.0 (0.0, 15.0)	0.0 (0.0, 15.0)	<0.001
Neostigmine	0.0 (0.0, 1.0)	0.0 (0.0, 1.0)	0.026
Sugammadex	0.0 (0.0, 0.0)	0.0 (0.0, 0.0)	<0.001
Fentanyl	0.0 (0.0, 0.1)	0.0 (0.0, 0.0)	<0.001
Alfentanil	0.0 (0.0, 0.0)	0.0 (0.0, 0.0)	0.171
Sufentanil	0.0 (0.0, 0.0)	0.0 (0.0, 0.0)	<0.001
Remifentanil	0.0 (0.0, 1.0)	0.0 (0.0, 1.0)	<0.001
Pethidine	0.0 (0.0, 0.0)	0.0 (0.0, 0.0)	<0.001
BUN	13.3 (10.5, 16.6)	16.1 (12.1, 21.6)	<0.001
Cr	0.8 (0.6, 0.9)	0.8 (0.7, 1.1)	<0.001
GFR	98.0 (83.0, 116.0)	85.0 (63.0, 107.0)	<0.001
PT	12.1 (11.3, 12.9)	12.9 (12.0, 13.9)	<0.001
aPTT	31.6 (27.9, 35.2)	33.2 (28.5, 37.8)	<0.001
INR	1.0 (0.9, 1.1)	1.0 (1.0, 1.1)	<0.001
PLT	246.0 (206.0, 291.0)	219.0 (171.0, 276.0)	<0.001
Na	140.0 (138.0, 141.0)	139.0 (136.0, 141.0)	<0.001
K	4.1 (3.9, 4.4)	4.0 (3.7, 4.3)	<0.001
Uric acid	4.7 (3.7, 5.8)	4.6 (3.4, 5.9)	0.004
Protein	7.2 (6.7, 7.5)	6.6 (6.0, 7.2)	<0.001
Albumin	4.4 (4.1, 4.6)	3.9 (3.3, 4.3)	<0.001
Robotic surgery	5022 (2.3)	117 (3.3)	<0.001
Laparoscopic surgery	41,921 (19.2)	436 (12.2)	<0.001
Heart surgery	970 (0.4)	265 (7.4)	<0.001
Abdominal surgery (minor/major)	34,286 (15.7)/6750 (3.1)	434 (12.1)/369 (10.3)	<0.001
Breast surgery (minor/major)	7304 (3.4)/20 (0.0)	19 (0.5)/2 (0.1)	<0.001
Ear surgery (minor/major)	4270 (2.0)/2 (0.0)	8 (0.2)/0 (0)	<0.001
Endocrine surgery (minor/major)	3145 (1.4)/2691 (1.2)	19 (0.5)/7 (0.2)	<0.001
Eye surgery	4215 (1.9)	10 (0.3)	<0.001
Head and neck surgery (minor/major)	25,006 (11.4)/269 (0.1)	41 (1.1)/8 (0.2)	<0.001
Musculoskeletal surgery (minor/major)	50,412 (23.1)/3040 (1.4)	666 (18.6)/306 (8.5)	<0.001
Neurosurgery (minor/major)	5837 (2.7)/1707 (0.8)	306 (8.5)/197 (5.5)	<0.001
OBGY surgery (minor/major)	33,713 (15.4)/602 (0.3)	128 (3.6)/30 (0.8)	<0.001
Spine surgery (minor/major)	4181 (1.9)/2743 (1.3)	144 (4.0)/208 (5.8)	<0.001
Thoracic surgery (minor/major)	3130 (1.4)/945 (0.4)	142 (4.0)/71 (2.0)	<0.001
Transplantation surgery (minor/major)	64 (0.0)/127 (0.1)	4 (0.1)/37 (1.0)	<0.001
Urogenital surgery (minor/major)	19,458 (8.9)/1894 (0.9)	98 (2.7)/118 (3.3)	<0.001
Vascular surgery (minor/major)	1610 (0.7)/82 (0.0)	76 (2.1)/25 (0.7)	<0.001
Skin and soft tissue surgery (minor/major)	15,406 (7.1)/97 (0.0)	111 (3.1)/7 (0.2)	<0.001

AIDS, acquired immunodeficiency syndrome; aPTT, activated partial thromboplastin time; ARDS, acute respiratory distress; ASA-PS, American Society of Anesthesiologists physical status; BUN, blood urea nitrogen; Cr, creatine; FFP, fresh frozen plasma, GFR, glomerular filtration rate; HIV, human immunodeficiency virus; INR, international normalized ratio; OBGY, obstetric and gynecologic; PC, platelet concentrate; PLT, platelet; PPE, postoperative pulmonary edema; PT, prothrombin time; PTE, pulmonary thromboembolism.

**Table 2 jcm-12-01804-t002:** Test dataset.

	No PPE (*n* = 33,095)	PPE (*n* = 1896)	*p*-Value
Age, year	52.0 (39.0, 64.0)	71.0 (60.0, 79.0)	<0.001
Male sex	18,038 (54.5)	973 (51.3)	0.007
Order of surgery	1.0 (1.0, 1.0)	1.0 (1.0, 1.0)	<0.001
Preoperative atelectasis	788 (2.4)	172 (9.1)	<0.001
Preoperative effusion	725 (2.2)	194 (10.2)	<0.001
Preoperative pneumothorax	191 (0.6)	25 (1.3)	<0.001
Preoperative pneumonia	257 (0.8)	109 (5.8)	<0.001
Preoperative PTE	17 (0.1)	6 (0.3)	<0.001
Preoperative ARDS	6 (0.0)	0 (0)	>0.999
Body mass index	24.4 (22.1, 26.9)	23.9 (21.3, 26.5)	<0.001
Congestive heart failure	1168 (3.5)	238 (12.6)	<0.001
Cardiac arrhythmias	1438 (4.3)	231 (12.2)	<0.001
Valvular disease	231 (0.7)	51 (2.7)	<0.001
Pulmonary circulation disorders	239 (0.7)	65 (3.4)	<0.001
Peripheral vascular disorders	534 (1.6)	73 (3.9)	<0.001
Hypertension, uncomplicated	3896 (11.8)	467 (24.6)	<0.001
Hypertension, complicated	2507 (7.6)	264 (13.9)	<0.001
Paralysis	282 (0.8)	38 (2.0)	<0.001
Other neurological disorders	1021 (3.1)	163 (8.6)	<0.001
Chronic pulmonary diseases	4377 (13.2)	380 (20.0)	<0.001
Diabetes, uncomplicated	3325 (10.1)	338 (17.8)	<0.001
Diabetes, complicated	2067 (6.2)	251 (13.2)	<0.001
Hypothyroidism	537 (1.6)	54 (2.9)	<0.001
Renal failure	1378 (4.2)	236 (12.4)	<0.001
Liver disease	2556 (7.7)	221 (11.7)	<0.001
Peptic ulcer diseases (excluding bleeding)	1147 (3.5)	75 (4.0)	0.287
AIDS/HIV	2 (0.0)	0 (0)	>0.999
Lymphoma	132 (0.4)	9 (0.5)	0.749
Metastatic cancer	333 (1.0)	57 (3.0)	<0.001
Solid tumors without metastasis	4336 (13.1)	595 (31.4)	<0.001
Rheumatoid arthritis/collagen vascular diseases	655 (2.0)	45 (2.4)	0.268
Alcohol consumption	8651 (26.1)	309 (16.3)	<0.001
Current smoking	5953 (18.0)	258 (13.6)	<0.001
Smoking frequency (packs)	0.0 (0.0, 0.0)	0.0 (0.0, 0.0)	<0.001
Smoking duration (years)	0.0 (0.0, 0.0)	0.0 (0.0, 0.0)	<0.001
Emergency	4606 (13.9)	596 (31.4)	<0.001
ASA-PS > 2	13,558 (41.0)	829 (2.5)	<0.001
General anesthesia	29,306 (88.5)	1886 (99.5)	<0.001
Anesthetics (Sevoflurane)	16,024 (48.4)	888 (2.7)	<0.001
N_2_O	18,445 (55.7)	578 (30.5)	<0.001
Anesthesia time (min)	85.0 (60.0, 130.0)	140.0 (95.0, 215.0)	<0.001
Surgery time (min)	55.0 (35.0, 95.0)	100.0 (60.0, 165.0)	<0.001
Intraoperative blood administration	0.0 (0.0, 0.0)	0.0 (0.0, 240.0)	<0.001
Intraoperative fluid administration	350.0 (200.0, 600.0)	950.0 (500.0, 1750.0)	<0.001
Intraoperative urine output	0.0 (0.0, 30.0)	85.0 (20.0, 230.0)	<0.001
Estimated blood loss	20.0 (5.0, 50.0)	200.0 (30.0, 600.0)	<0.001
Arterial line	7387 (22.3)	1627 (85.8)	<0.001
Central venous line	3314 (10.0)	1258 (66.3)	<0.001
Foley catheter	10,807 (32.6)	1560 (82.3)	<0.001
Levin tube	1070 (3.2)	278 (14.7)	<0.001
Patient-controlled analgesia (intravenous)	15,263 (46.1)	490 (1.5)	<0.001
Intraoperative packed red blood cell	0.0 (0.0, 0.0)	0.0 (0.0, 1.0)	<0.001
Intraoperative FFP	0.0 (0.0, 0.0)	0.0 (0.0, 0.0)	<0.001
Intraoperative PC	0.0 (0.0, 0.0)	0.0 (0.0, 0.0)	<0.001
Intraoperative cryoprecipitate	0.0 (0.0, 0.0)	0.0 (0.0, 0.0)	<0.001
Rocuronium	50.0 (50.0, 75.0)	75.0 (50.0, 150.0)	<0.001
Vecuronium	0.0 (0.0, 0.0)	0.0 (0.0, 0.0)	>0.999
Atracurium	0.0 (0.0, 0.0)	0.0 (0.0, 0.0)	<0.001
Cisatracurium	0.0 (0.0, 0.0)	0.0 (0.0, 0.0)	<0.001
Succinylcholine	0.0 (0.0, 0.0)	0.0 (0.0, 0.0)	0.081
Pyridostigmine	20.0 (0.0, 20.0)	0.0 (0.0, 20.0)	<0.001
Neostigmine	0.0 (0.0, 0.0)	0.0 (0.0, 0.0)	>0.999
Sugammadex	0.0 (0.0, 0.0)	0.0 (0.0, 200.0)	<0.001
Fentanyl	0.0 (0.0, 0.0)	0.0 (0.0, 0.0)	0.056
Alfentanil	0.0 (0.0, 0.5)	0.0 (0.0, 0.2)	<0.001
Sufentanil	0.0 (0.0, 0.0)	0.0 (0.0, 0.0)	0.095
Remifentanil	0.0 (0.0, 0.0)	0.0 (0.0, 0.4)	<0.001
Pethidine	0.0 (0.0, 0.0)	0.0 (0.0, 0.0)	<0.001
BUN	14.1 (11.5, 17.4)	16.4 (12.6, 22.0)	<0.001
Cr	0.8 (0.7, 1.0)	0.9 (0.7, 1.1)	0.002
GFR	90.0 (77.0, 104.0)	81.6 (63.0, 100.7)	<0.001
PT	11.1 (10.6, 11.7)	11.6 (11.0, 12.5)	<0.001
aPTT	31.2 (29.0, 33.7)	30.0 (27.5, 32.8)	<0.001
INR	1.0 (1.0, 1.1)	1.1 (1.0, 1.1)	<0.001
PLT	247.0 (208.0, 292.0)	218.0 (169.8, 274.0)	<0.001
Na	141.0 (140.0, 143.0)	140.0 (138.0, 142.0)	<0.001
K	4.2 (3.9, 4.4)	4.1 (3.8, 4.4)	<0.001
Uric acid	4.8 (3.8, 5.9)	4.5 (3.4, 5.8)	<0.001
Protein	7.0 (6.6, 7.3)	6.5 (6.0, 7.0)	<0.001
Albumin	4.4 (4.1, 4.6)	3.9 (3.4, 4.3)	<0.001
Robotic surgery	208 (0.6)	73 (3.9)	<0.001
Laparoscopic surgery	4240 (12.8)	427 (22.5)	<0.001
Heart surgery	24 (0.1)	8 (0.4)	<0.001
Abdominal surgery (minor/major)	5113 (15.4)/706 (2.1)	428 (22.6)/268 (14.1)	<0.001
Breast surgery (minor/major)	1306 (4.0)/2 (0.0)	5 (0.3)/1 (0.1)	<0.001
Ear surgery (minor/major)	855 (2.6)	1 (0.1)	<0.001
Endocrine surgery (minor/major)	369 (1.1)/107 (0.3)	8 (0.4)/1 (0.1)	0.002
Eye surgery	606 (1.8)	4 (0.2)	<0.001
Head and neck surgery (minor/major)	4087 (12.3)/12 (0.0)	22 (1.2)/3 (0.2)	<0.001
Musculoskeletal surgery (minor/major)	11,044 (33.4)/530 (1.6)	353 (18.6)/226 (11.9)	<0.001
Neurosurgery (minor/major)	779 (2.4)/231 (0.7)	99 (5.2)/96 (5.1)	<0.001
OBGY surgery (minor/major)	1705 (5.2)/24 (0.1)	5 (0.3)/1 (0.1)	<0.001
Spine surgery (minor/major)	2050 (6.2)/220 (0.7)	73 (3.9)/44 (2.3)	<0.001
Thoracic surgery (minor/major)	272 (0.8)/54 (0.2)	76 (4.0)/19 (1.0)	<0.001
Transplantation surgery (minor/major)	0 (0)	0 (0)	>0.999
Urogenital surgery (minor/major)	2021 (6.1)/104 (0.3)	62 (3.3)/69 (3.6)	<0.001
Vascular surgery (minor/major)	368 (1.1)/2 (0.0)	13 (0.7)/0 (0)	0.208
Skin and soft tissue surgery (minor/major)	1510 (4.6)/14 (0.0)	12 (0.6)/4 (0.2)	<0.001

AIDS, acquired immunodeficiency syndrome; aPTT, activated partial thromboplastin time; ARDS, acute respiratory distress; ASA-PS, American Society of Anesthesiologists physical status; BUN, blood urea nitrogen; Cr, creatine; FFP, fresh frozen plasma, GFR, glomerular filtration rate; HIV, human immunodeficiency virus; INR, international normalized ratio; OBGY, obstetric and gynecologic; PC, platelet concentrate; PLT, platelet; PPE, postoperative pulmonary edema; PT, prothrombin time; PTE, pulmonary thromboembolism.

**Table 3 jcm-12-01804-t003:** Best threshold, precision, recall, f1 score, and accuracy values for each model.

	Best Threshold	Accuracy		Precision	Recall	F1 Score
BRF	0.42	0.82	Normal	0.99	0.81	0.89
			PPE	0.21	0.89	0.34
LGBM	0.048	0.72	Normal	0.98	0.72	0.83
			PPE	0.14	0.80	0.24
XGB	0.353	0.76	Normal	0.98	0.77	0.86
			PPE	0.14	0.66	0.23
MLP	0	0.93	Normal	0.95	0.98	0.96
			PPE	0.07	0.03	0.04
LR	697.8	0.75	Normal	0.98	0.75	0.85
			PPE	0.14	0.68	0.23

BRF, balanced random forest; LGBM, light-gradient boosting machine; XGB, extreme-gradient boosting machine; MLP, multilayer perceptron; LR, logistic regression; PPE, postoperative edema.

**Table 4 jcm-12-01804-t004:** Validation of the under-sampling test dataset and simplified model using balanced random forest algorithm.

	AUC (95% CI)	Best Threshold	Accuracy		Precision	Recall	F1 Score
After under sampling	0.911 (0.855–0.972)	0.42	0.83	Normal	0.99	0.82	0.90
				PPE	0.22	0.89	0.36
Simplified model	0.901 (0.829–0.978)	0.44	0.82	Normal	0.99	0.81	0.89
				PPE	0.21	0.88	0.34

AUC, area under curve; CI, confidence interval; PPE, postoperative pulmonary edema.

## Data Availability

Restrictions apply to the availability of these data. Data was obtained from Hallym Medical Center and are available from clinical data warehouse of Hallym Medical Center with the permission of Hallym Medical Center.

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
