# Peer review of "Prediction of Postoperative Pulmonary Edema Risk Using Machine Learning"

_jcm, 2023, doi:10.3390/jcm12051804_

Round 1

Reviewer 1 Report

Plentiful previous studies have developed ML models to predict postoperative pulmonary complications. But they neither predicted PPE risk after all type of surgery nor specially set predicting PPE risk as their purpose. In this paper, they are the first to develop models to predict PPE risk exclusively under-going surgery based on five ML models. A large, multi-central dataset was collected, and 98 perioperative variables were taken into account, confirming the accuracy and generalization of the models. By comparing the results of all ML models and choosing the best among them, their final model is superior to previously reported prediction models.

However, this article still has some deficiencies as follows:

1.        The background of the study is inadequate. Relevant research background needs to be supplemented in INTRODUCTION. And they should cite all papers you use properly.

2.        A flow chart is suggested to be provided in the paper to show the process of data filtering and data cleansing during the collection of your dataset.

3.        As their dataset is large and complex, a validation dataset can be carved out additionally to validate the performance of their models so that they can modify the models to predict more accurately.

4.        The clinical significance of their study needs to be further deepened. As the best predictive model is found, an online calculator can be built based on this method to calculate the PEE risk, so that it can be applied clinically.

Author Response

Response to Reviewer 1.

Please find attached a revised version of our manuscript, “Prediction of postoperative pulmonary edema risk using ma-chine learning” (jcm-2218428).

We thank you for your thoughtful suggestions regarding the original version of our paper; most of the suggested changes have been incorporated into the revision.

All revisions are described in detail in the order mentioned in the review, following your comment. We believe that the revisions have greatly improved the manuscript and hereby submit the revised version for your consideration for publication.

  1. Comment:

The background of the study is inadequate. Relevant research background needs to be supplemented in INTRODUCTION. And they should cite all papers you use properly.

Answer:

We thank the reviewer for these comments and suggestions, which have improved our manuscript.

We have added text and reference to the Introduction.

  1. Introduction

Postoperative pulmonary edema (PPE) is a well-known complication with multiple possible causes [1]. Preexisting cardiac disease, including heart failure, is the most common cause of PPE. Fluid overload results in increased hydrostatic pressure and worsening left ventricular function [2]. Regardless of preexisting heart disease, fluid overload itself can cause PPE. In particular, excessive postoperative fluid administration and transfusions increase the risk of PPE [1,3]. Neurogenic pulmonary edema is another potential cause of PPE [4]. Although neurogenic pulmonary edema is sometimes regarded as a form of acute respiratory distress syndrome, its pathophysiology and prognosis differ from the characteristics of acute respiratory distress [4,5]. PPE can also be caused by anaphylaxis, which results in negative pressure and acute lung injury [1,6].

It is often difficult to determine the cause of PPE during its early stages, particularly in patients with overlapping etiologies [1,6,7]. There is a need to identify patients at high risk for PPE to allow prevention and early treatment. Several studies have reported the causes and risk factors for PPE, but early diagnosis and management are difficult, more so in patients with overlapping etiologies or uncertain causes [1-10].

Advances in computing have enhanced several key areas of clinical research; artificial intelligence-based methods may have additional applications. Machine learning (ML) systems are widely used in clinical research to analyze big data. Compared to traditional scoring systems, ML models perform better in predicting various clinical conditions [11-13]. They have been successfully used to predict postoperative complications [14-19]. However, there is no reported ML model to predict PPE. In the present study, we hypothesized that ML could predict PPE risk with good performance, then developed ML models to predict PPE.

  1. Comment:

A flow chart is suggested to be provided in the paper to show the process of data filtering and data cleansing during the collection of your dataset.

Answer:

We have also added the flow chart in Figure 1.

Figure 1. Data processing and machine learning process

LGBM, light-gradient boosting machine; LR, logistic regression; ML, machine learning; MLP, multilayer perceptron; XGB, extreme-gradient boosting machine

  1. Comment:

As their dataset is large and complex, a validation dataset can be carved out additionally to validate the performance of their models so that they can modify the models to predict more accurately.

Answer:

Thank you for your good advice. We made additionally modified dataset and prediction model, and validated with them.

2.5 Modified dataset

An additional carved dataset was used to modify the prediction model based on the large and complex dataset. This dataset was learned and validated using the best prediction algorithm from the original data. First, the test dataset was reduced by under-sampling using the Tomek’s link method to validate our best model [30]. Second, a simplified prediction model was made using 20 important features of the best model, and was validated using a test dataset that included these features. 

Table 4. Validation of the under-sampling test dataset and simplified model using balanced random forest algorithm

AUC

(95% CI)

Best

threshold

Accuracy

Precision

Recall

F1 score

After under sampling

0.911

(0.855 – 0.972)

0.42

0.83

Normal

0.99

0.82

0.90

PPE

0.22

0.89

0.36

Simplified model

0.901

(0.829 – 0.978)

0.44

0.82

Normal

0.99

0.81

0.89

PPE

0.21

0.88

0.34

AUC, area under curve; CI, confidence interval; PPE, postoperative pulmonary edema

  1. Comment:

The clinical significance of their study needs to be further deepened. As the best predictive model is found, an online calculator can be built based on this method to calculate the PEE risk, so that it can be applied clinically.

       Answer:

Thank you for your very good opinion. We will accept your opinion and install a more practical online calculator that predicts PPE using our prediction model in the future.

Reviewer 2 Report

The study aimed to evaluate the potential of machine learning algorithms in predicting the risk of postoperative pulmonary edema (PPE) using pre- and intraoperative data. The authors analyzed medical records of patients aged over 18 years who underwent surgery in five South Korean hospitals between 2011 and 2021. The data from four of the hospitals was used as the training dataset, while data from the remaining hospital was used as the test dataset. Five machine learning algorithms were used including extreme gradient boosting, light gradient boosting machine, multilayer perceptron, logistic regression, and balanced random forest (BRF). The performance of the models was evaluated using various metrics such as area under the ROC curve, precision, recall, f1 score, and accuracy. The results showed that the BRF model had the best performance with an area under the ROC curve of 0.91. The five most important features for PPE prediction were arterial line monitoring, American Society of Anesthesiologists physical status, urine output, age, and Foley catheter status. The study concluded that machine learning models, such as BRF, could predict PPE risk and improve postoperative management.

The manuscript can be improved from the following aspects:

1.     English writing needs to be improved by native speakers.

2.     The conclusion needs to make more sense. If AUC-ROC is high for the BRF model, but its precision and f1 score metrics are not good. That means authors need to adjust the decision threshold finding by AUC-ROC.

3.     The discussion should mention the study’s limitations and future work.

4.     The clinical connection with significant variables should be discussed as well.

Author Response

Please find attached a revised version of our manuscript, “Prediction of postoperative pulmonary edema risk using ma-chine learning” (jcm-2218428).

We thank you for your thoughtful suggestions regarding the original version of our paper; most of the suggested changes have been incorporated into the revision.

All revisions are described in detail in the order mentioned in the review, following your comment. We believe that the revisions have greatly improved the manuscript and hereby submit the revised version for your consideration for publication.

The study aimed to evaluate the potential of machine learning algorithms in predicting the risk of postoperative pulmonary edema (PPE) using pre- and intraoperative data. The authors analyzed medical records of patients aged over 18 years who underwent surgery in five South Korean hospitals between 2011 and 2021. The data from four of the hospitals was used as the training dataset, while data from the remaining hospital was used as the test dataset. Five machine learning algorithms were used including extreme gradient boosting, light gradient boosting machine, multilayer perceptron, logistic regression, and balanced random forest (BRF). The performance of the models was evaluated using various metrics such as area under the ROC curve, precision, recall, f1 score, and accuracy. The results showed that the BRF model had the best performance with an area under the ROC curve of 0.91. The five most important features for PPE prediction were arterial line monitoring, American Society of Anesthesiologists physical status, urine output, age, and Foley catheter status. The study concluded that machine learning models, such as BRF, could predict PPE risk and improve postoperative management.

The manuscript can be improved from the following aspects:

  1. Comment:

English writing needs to be improved by native speakers.

Answer:

Thank you for your comments. Our manuscript was edited in English by native speakers.

  1. Comment:

The conclusion needs to make more sense. If AUC-ROC is high for the BRF model, but its precision and f1 score metrics are not good. That means authors need to adjust the decision threshold finding by AUC-ROC.

Answer:

Thank you for your comments. We found the best threshold and summarized the Precision, Recall, f1 score and Accuracy accordingly in Table 3.

Table 3. Best threshold, precision, recall, f1 score, and accuracy values for each model

Best threshold

Accuracy

Precision

Recall

f1 score

BRF

0.42

0.82

Normal

0.99

0.81

0.89

PPE

0.21

0.89

0.34

LGBM

0.048

0.72

Normal

0.98

0.72

0.83

PPE

0.14

0.80

0.24

XGB

0.353

0.76

Normal

0.98

0.77

0.86

PPE

0.14

0.66

0.23

MLP

0

0.93

Normal

0.95

0.98

0.96

PPE

0.07

0.03

0.04

LR

697.8

0.75

Normal

0.98

0.75

0.85

PPE

0.14

0.68

0.23

BRF, balanced random forest; LGBM, light-gradient boosting machine; XGB, extreme-gradient boosting machine; MLP, multilayer perceptron; LR, logistic regression; PPE, postoperative edema

  1. Comment:

The discussion should mention the study’s limitations and future work.

Answer:

Thank you for your comments. We added study's limitations and future work in the discussion.

To the best of our knowledge, our model is the first to predict PPE risk, and its perfor-mance was better than the previous PPE models. However, the present study had some limitations. First, the overall performance of the model was good, but its precision and f1 score were low, even for the best threshold. Because recall (sensitivity) was good, the pro-portion of false positives may be high, presumably because of the low proportion of pa-tients with pulmonary edema in the overall dataset. Thus, our model interpreted the nor-mal state as PPE in many cases. There were similar results in the validation with the un-der-sampling dataset. Second, our model requires many features to predict PPE, which reduces its practicality. Although the performance was not significantly worse in model with twenty features, this limitation of the model could not be resolved. A prediction model based on fewer features while maintaining the performance may be needed in the future. To resolve the first two limitations, additional datasets should be acquired and learned, or features with better predictive values should be selected. Third, our model could not distinguish between cardiogenic and noncardiogenic PPE. Additional studies are needed to develop models that can distinguish between the two PPE types and predict the risk for each type.

  1. Comment:

The clinical connection with significant variables should be discussed as well.

Answer:

Thank you for good advice. We added the clinical connection with significant variables.

The most important feature was arterial line monitoring, which is required in patients who need continuous blood pressure monitoring or multiple blood sampling during sur-gery [32]. Arterial line monitoring is the standard of care for patients at risk for rapid he-modynamic changes. Patients with a poor preoperative status and those who undergo major surgeries can develop rapid hemodynamic changes and often need multiple sam-pling [33]. American Society of Anesthesiologists physical status and age also indicate preoperative patient condition. Patients with high American Society of Anesthesiologists physical status grades may develop heart, lung, kidney, and brain problems [34,35]. Old age is generally associated with compromised organ function, resulting in a greater risk for PPE [36]. Urine output, fluid volume, EBL, albumin, and glomerular filtration rate di-rectly and indirectly affect body fluid status, which is associated with hydrostatic pressure [37-41].
